# Improving the Therapeutic Index of Smp24, a Venom-Derived Antimicrobial Peptide: Increased Activity against Gram-Negative Bacteria

**DOI:** 10.3390/ijms23147979

**Published:** 2022-07-20

**Authors:** Kirstie M. Rawson, Melissa M. Lacey, Peter N. Strong, Keith Miller

**Affiliations:** Biomolecular Sciences Research Centre, Department of Biosciences and Chemistry, Sheffield Hallam University, Howard Street, Sheffield S1 1WB, UK; kirstie.rawson@btinternet.com (K.M.R.); m.lacey@shu.ac.uk (M.M.L.); p.strong@shu.ac.uk (P.N.S.)

**Keywords:** antimicrobial peptide, scorpion venom, peptide modification

## Abstract

Antimicrobial peptides (AMPs) are naturally occurring compounds which possess a rapid killing mechanism and low resistance potential. Consequently, they are being viewed as potential alternatives to traditional antibiotics. One of the major factors limiting further development of AMPs is off-target toxicity. Enhancements to antimicrobial peptides which can maximise antimicrobial activity whilst reducing mammalian cytotoxicity would make these peptides more attractive as future pharmaceuticals. We have previously characterised Smp24, an AMP derived from the venom of the scorpion *Scorpio maurus palmatus*. This study sought to better understand the relationship between the structure, function and bacterial selectivity of this peptide by performing single amino acid substitutions. The antimicrobial, haemolytic and cytotoxic activity of modified Smp24 peptides was determined. The results of these investigations were compared with the activity of native Smp24 to determine which modifications produced enhanced therapeutic indices. The structure–function relationship of Smp24 was investigated by performing N-terminal, mid-chain and C-terminal amino acid substitutions and determining the effect that they had on the antimicrobial and cytotoxic activity of the peptide. Increased charge at the N-, mid- and C-termini of the peptide resulted in increased antimicrobial activity. Increased hydrophobicity at the N-terminus resulted in reduced haemolysis and cytotoxicity. Reduced antimicrobial, haemolytic and cytotoxic activity was observed by increased hydrophobicity at the mid-chain. Functional improvements have been made to modified peptides when compared with native Smp24, which has produced peptides with enhanced therapeutic indices.

## 1. Introduction

Antibiotics have been the primary treatment option for bacterial infection since their introduction in the 1940s and have saved millions of lives worldwide [1]. Unfortunately, the availability of short-course drugs at a low cost in the 1940’s set the bar for an unsustainable future, which has led to an increase in drug-resistant pathogens and disinvestment from the pharmaceutical industry [2]. New antimicrobial molecules with alternative modes of action are required if increasing global antibiotic resistance is to be combatted [3]. Antimicrobial peptides (AMPs) are evolutionarily conserved components of the innate immune system which serve to protect the host from invading pathogens and infection [4]. AMPs have rapidly captured growing interest as potential drug candidates [5], particularly because their mechanism of actions suggests a reduced ability of the bacteria to develop resistance. The presence of AMPs has been reported across all kingdoms of life (bacteria, archaea, protists, fungi, plants and animals) with examples including biocins from bacteria [6]; halocins from haloarchaea [7]; amoebapores from *Entamoeba histolytica* [8]; CSαβ-type defensins [9] and BP plectasin [10] from fungi; thionins, defensins and hevein-like peptides from plants [11]; and defensins, cathelicidins and histatins from animals, including humans [12]. These animal-derived antimicrobials include AMPs from the venoms of scorpions, where AMPs are thought to protect the animal from exogenous infection following interaction with their prey [13].

Despite the structural and functional diversity of AMPs, most share common physicochemical properties [14]. AMPs are small (typically 12–60 amino acids), amphipathic molecules with a net positive charge (+2 to +9) and approximately 50% hydrophobic content [4,8]. They demonstrate a multitude of functions, but their main mode of action is bactericidal, which is often achieved by selective disruption of prokaryotic membranes [15]. One of the most important hurdles that must be overcome in the future development of AMPs is providing a drug with a viable therapeutic index (TI). The ability to specifically target prokaryotic membranes with a sufficient degree of selectivity with respect to mammalian eukaryotic membranes is crucial in this regard and has proved extremely difficult. However, a few notable successes have been achieved, including nisin, which has strong antimicrobial activity against Gram-positive organisms [16], and polymyxins, which are used as the last-resort option for the treatment of drug-resistant Gram-negative infections [5]. Many other AMPs are undergoing clinical trial assessment, which provides optimism for the introduction of novel AMP-based drugs as potential future antimicrobials [5].

Scorpion venom is a rich source of amphipathic AMPs [13]. Smp24 is a 24-residue antimicrobial peptide derived from the venom of the scorpion *Scorpio maurus palmatus* [17]. It belongs to a family of scorpion venom AMPs without disulphide bridges [13] which have been particularly useful in studying the mechanisms of peptide interaction with model phospholipid membranes. Smp24 shares 54% sequence homology with another family member, pandinin 2 (pin2) [17], and adopts an α-helical structure in membrane-mimicking environments. It carries a charge of +3 and has broad-spectrum antimicrobial activity [18]. We have previously utilised AMPs from *S. maurus palmatus* (Smp24 and Smp43) to study the mechanisms of peptide interaction with model phospholipid membranes using a variety of techniques and have visualised a diffusion-limited membrane attack of Smp43 using atomic force microscopy [19]. We have also carried out preliminary structure–function studies on Smp24 by inserting a structural GVG hinge into the centre of the peptide, thereby breaking the alpha-helical structure into two allowing us to investigate the role of the C-terminus [17]. This present study seeks to expand that investigation and improve our understanding of the relationship between structure and function of Smp24 by investigating the effects of charge and hydrophobicity on both prokaryotic and eukaryotic activity. A systematic approach has been taken to increase the net charge of the peptide, as this has previously been shown to increase antimicrobial activity [20]. We have also increased the hydrophobicity of Smp24 at both the N- and C-termini as well as in the centre of the peptide, again in the light of evidence from previous studies [4,14]. Using this strategy, we have provided insights into the structure–activity relationships of scorpion venom AMPs with the aim of increasing their TI.

## 2. Results

A series of ten peptide derivatives of Smp24 were synthesised: nine derivatives possessing one amino acid substitution and one derivative possessing two substitutions (Table 1). Residues were selected for substitution based on analysis of the physicochemical properties of different point substitutions of Smp24 (https://web.expasy.org/protparam/, accessed on 16 June 2022).

Antimicrobial activity (minimum inhibitory concentration (MIC) and minimum bactericidal concentration (MBC) values) against *Escherichia coli*, *Pseudomonas aeruginosa* and *Staphylococcus aureus* was determined for each of the ten peptides as well as for daptomycin and polymyxin B as representative, commercially available AMPs. Smp24 has three positive charges, and all peptides with an increased charge (up to +5) displayed increased levels of antimicrobial activity (Table 2). In contrast, decreasing the charge (to +2) reduced activity.

No antimicrobial activity was detected for Smp24 K7F (up to 512 μg/mL) against any of the bacterial isolates tested. By contrast, S3K, S15K and S24K modifications yielded peptides with the highest activity against Gram-positive organisms (Table 2). The MIC and MBC values obtained from these modifications for *S. aureus* were lower than for polymyxin B and were comparable with the activity of daptomycin via broth microdilution [22]. In addition to inhibiting the growth of bacteria, the S15K and S24K modifications demonstrated bactericidal activity comparable to that of daptomycin, with the S3K modification demonstrating a four-fold increase in bactericidal activity compared with the parental peptide. Similarly, the S3K, S15K and S24K modifications exhibited enhanced activity against *E. coli*, with the S3K modification demonstrating greater antimicrobial activity than that of polymyxin B. The activity of the peptides against *P. aeruginosa* was mixed, with the S15K/S24K double modification demonstrating a four-fold increase in activity compared with parental Smp24. Unlike the effects of charge, peptide derivatives with increasing hydrophobicity (as shown by GRAVY values) showed limited changes (up to a two-fold decrease) in antimicrobial activity.

Smp24 demonstrated significant erythrocyte disruption in haemolysis assays, with an HC_50_ concentration of 76 μg/mL (Table 3). Increasing the charge of Smp24 resulted in peptides with enhanced haemolytic activity, with the S3K modification being the most haemolytic of all the modifications performed. Increasing the hydrophobicity of the peptides resulted in peptides with reduced haemolytic activity. The K7F, W2A and F4A modifications demonstrated statistically significant decreases in haemolytic activity with HC_50_ values of 442 μg/mL, >512 μg/mL and 423 μg/mL, respectively. Of all the modified peptides in this study, only these three peptides had values comparable with daptomycin or polymyxin B (HC_50_ values ≧ 512 μg/mL). Although there are differences between the composition of sheep and human erythrocytes (the former having a higher percentage of phosphatidylserine and sphingomyelin), they share sufficient commonalities to be a suitable first screen for cytotoxicity [23,24].

Smp24 demonstrated a high level of cytotoxicity against HepG2 and HEK293 cells (LD_50_ of 37 and 39 μg/mL respectively) (Table 4). Modifications to the structure of Smp24 did not result in any peptides with a statistically significant increase in cytotoxicity against HepG2 cells, and five of the peptides—K7F, D23F, S3F, W2A and F4A—demonstrated a significant decrease in cytotoxic activity, with K7F and D23F modifications both demonstrating LD_50_ values above 512 μg/mL, a level which is comparable with those of daptomycin and polymyxin B. In contrast, the cytotoxic effects of Smp24 peptides on HEK293 cells were variable. S15K, S24K, S3K/S15K and K7S demonstrated significantly increased LD_50_ values, although K7F, W2A and F4A modifications showed significant reductions in cytotoxicity, with K7F demonstrating a lower cytotoxicity profile than polymyxin B.

TI is a widely accepted parameter to represent the specificity of antimicrobial compounds for prokaryotic cells with respect to eukaryotic cells [25,26]. Higher values demonstrate greater specificity for prokaryotic cells. Significant improvements were observed for the TI of modified Smp24 analogues (Figure 1).

S3K, S15K, S24K, F4A and W2A modifications all showed statistically significant improvements in TI as compared with the parent peptide, although K7F and K7S modifications showed significant decreases. Some of the modifications also resulted in significant decreases in TI, such as K7F and K7S. Interestingly, the D23F modification produced a significantly improved TI when calculated against liver cells, but the peptide had a decreased TI against erythrocytes, which highlights the importance of full-panel cytotoxicity testing. The most and least reactive modified peptides (S3K and K7F, respectively) were chosen for secondary structural analysis using circular dichroism (CD) spectroscopy [27] to determine if the single amino acid substitutions yielded detectable structural changes (Figure 2).

Observable differences were seen between Smp24 and most of the modified peptides. The peptides were disordered in aqueous solution and adopted an alpha-helical conformation which was characterised by the presence of two minima around 208 and 222 nm in the presence of 60% tetrafluoroethylene. While Smp24 S3K shares a similar CD structure to parental Smp24, Smp24 K7F has a less pronounced random coil structure in H_2_O and also adopts a less pronounced alpha-helical structure (as evidenced by less pronounced minima at 208 nm and 220 nm) in the presence of 60% TFE when compared with both Smp24 and Smp24 S3K. The effects of titrating TFE (0–60%) on the secondary structure of Smp24 and the S3K and K7F variants can be seen in Figure 3.

In our present understanding, the change from a disordered to an ordered alpha-helical conformation is a crucial part of the mechanism by which AMPs interact with biological membranes. Parental Smp24 adopts an α-helical structure in 25% TFE; Smp24 K7F adopts α-helical conformation at a lower concentration of TFE (20%), considerably higher concentrations of TFE (30%) are needed in order for Smp24 S3K to adopt an α-helical conformation.

Computational modelling was also used to investigate the relationship between peptide structure and antimicrobial activity. Helical wheel analysis showed that in Smp24, two distinct regions, one positively charged and one hydrophobic, were observed (Figure 4). Smp24 K7F replacement of a lysine residue with phenylalanine predicted a reduction in amphipathicity, reducing the charged domain on one side of the helix while not altering the hydrophobic domain on the opposite side (Figure 4). In contrast, in Smp24 SK3, incorporation of a lysine residue predicted an increase in peptide amphiphathicity; there was an increase in positive charge on one side of the helix, while the hydrophobic character of the opposite side of the helix remained unchanged (Figure 4).

PyMOL predicted an alpha-helical structure for Smp24, extending the length of the peptide. Introduction of a phenylalanine in Smp24 K7F introduced a significant disordered domain into the C-terminal region, and the alpha-helical structure seen in the parent peptide was reduced by almost 50% (Figure 5).

Introduction of an extra charged residue (Smp24 S3K) did not alter the helical nature of the parent peptide but increased the charge on one side of the helix while leaving the hydrophobic nature of the peptide on the other side of the helix unchanged. This can be most readily seen in space-filling models using the hydrophobic moment (HM) vector calculator and identifying polar and non-polar facets (Figure 6). This enables the evaluation of the 3D surface distribution of all hydrophilic and lipophilic regions of the peptide by calculating the electrostatic potential on the molecular surface based on atomic point charges [28].

Smp24 S3K shows greatly increased polarity at the N-terminus, and structural changes can be observed around the HM vector for Smp24 S3K with respect to the parent peptide, with other polar residues now appearing at the surface around the HM vector angle. There is an increase in the polar angle between the HM vector and the Z-axis hydrophobic moment from 127° to 137°. In contrast, the overall polarity of Smp24 K7F is reduced, with a reduction in the polar angle from 127° to 111°.

## 3. Discussion

One of the greatest challenges in developing new antimicrobials has been the lack of potency against Gram-negative bacterial pathogens [29]. It has previously been shown that increasing the charge at the N-terminus of recombinantly expressed defensins yielded the greatest increase in antimicrobial activity against both *E. coli* and *S. aureus* isolates [13,20]. Here, many modifications that resulted in an increased peptide charge increased the potency of Smp24 against both *E. coli* and *P. aeruginosa*, with the introduction of lysine at position 3 at the N-terminus (Smp24 S3K) producing the greatest increase in antimicrobial activity.

The main lipid components of bacterial cell membranes are anionic phospholipids, such as phosphatidylglycerol and cardiolipin. These anionic phospholipids, together with the predominant zwitterionic lipid phosphatidylethanolamine [30], cause the overall anionic charge across the bacterial membrane [26]. Increases in the positive charge of AMPs are associated with an increased electrostatic attraction to the membrane, which may explain the increase in potency of positively charged AMPs through pore formation and cell death [31]. Increasing the number of positive charges on Smp24 from +3 to +4 at either position 3 (S3K) or position 15 (S15K) produced an increase in antimicrobial activity in all isolates, both Gram-positive and Gram-negative. Conversely, reducing the charge from +3 to +2 (K7S, K7F) decreased the antimicrobial activity of all bacteria, both Gram-positive and Gram-negative. Interestingly, the increase to +5 charges (S3K/S15K), rather than increasing antibacterial action further as could be predicted by increased interaction with the bacterial membrane, in the doubly modified peptide Smp24 S3K/S15K was more potent against Gram-negative bacteria with respect to the parent peptide, whereas it was less potent against Gram-positive *S. aureus*. However, although the +5 double modification was more potent than the parent peptide, it was less potent than either of the +4 single mutants against Gram-negative bacteria, suggesting that simply increasing charge number was a simplistic concept in trying to increase biological activity. Increasing the charge on AMPs above a particular threshold is counterproductive due to electrostatic screening, [32,33,34] which may account for the antimicrobial profile of Smp24 S3K/S15K. In addition, different bacteria have subtly different membrane compositions, which is reflected in differences in cell membrane potential (see later), and this could provide both a rationale and a route for designing Gram-specific AMPs.

Hydrophobic modifications to Pandinin 2, a scorpion venom AMP from *Pandinus imperator* which shares 54% sequence homology with Smp24, increased the GRAVY score from 0.33 to 0.57, all yielded peptides with no detectible antimicrobial activity [35]. In this study, phenylalanine substitutions were made at the N-terminus, mid-peptide and C-terminus of Smp24 to investigate the effect of increasing hydrophobicity. The response was not straightforward; although most hydrophobic modifications of Smp24 (GRAVY scores 0.43 to 0.58) generally led to a modest two-fold reduction in antimicrobial activity, the most hydrophobic peptide in this study (K7F, GRAVY score 0.59), like the pandinin-2 derivatives, had no detectable antimicrobial activity. It is possible that at a critical level of hydrophobicity, peptides self-associate in an aqueous environment, decreasing their ability to insert into bacterial membranes. We have shown that increased antimicrobial activity of Smp24 S3K is associated with an increased HM angle. Similar results have been seen with other antimicrobial peptides. Charge modification of mastoparan (from +3 to +4), for example, increased the HM angle and enhanced antimicrobial activity [36]. Addition of lysine residues to Hp1404, an AMP from the scorpion *Hetrometrus petersii*, increased both antimicrobial activity and hydrophobic moment (Kim et al., 2018).

In order to assess the clinical usefulness of AMPs, it is necessary to compare peptide concentrations necessary disrupt eukaryotic membranes with those of bacterial membranes and thus determine the TI. The initial comparison made in most studies of membrane-lytic agents is to compare bacterial data with haemolytic data. However, data on nucleated mammalian cells in culture provide a more representative assessment of the likely impact of novel antimicrobials in terms of off-target effects. Here, we have examined the effects of Smp24 and derived peptides on sheep erythrocytes as well as liver (HepG2) and kidney (HEK293) cell lines, as liver and kidneys are major sites of drug accumulation in the body.

AMPs derived from many scorpion venoms have previously demonstrated a strong degree of haemolytic activity, with HC_50_ values often lower than or equal to the observed MIC [23,37]. The phenomenon was also observed here for Smp24. The haemolytic activity of Smp24 (HC_50_ = 75 μg/mL) also closely corresponded to the haemolytic activity of pandinin-2, which causes 51% lysis of sheep erythrocytes at a concentration of 57.5 μg/mL [13,23]. Smp24 derivatives showed a close correlation between HC_50_ values and antibacterial activity, with modified peptides that exhibited increased antibacterial activity (e.g., Smp24 S3K) being more haemolytic whereas peptides with decreased antibacterial activity (e.g., K7F) were less haemolytic.

Smp24 showed significant cytotoxicity against both HepG2 and HEK293 cells. None of the peptide modifications increased cytotoxicity against liver cells, although amino acid substitutions which incorporated phenylalanine residues into the structure resulted in a significant decrease in cytotoxicity (K7F, D23F, S3F). K7F and D23F both reduced cytotoxicity to below 512 µg/mL, comparable with lipopeptide antibiotics in clinical use. Contrastingly, the effect of peptide modification on kidney cells was more variable, with derivatives producing both increased and decreased levels of cytotoxicity. In contrast to what we observed in liver cells, modified peptides incorporating phenylalanine showed no decrease in kidney cell line cytotoxicity.

With the exception of Smp24 K7F, haemolytic (HC_50_) values do not correlate with the cytotoxicity data from mammalian cell lines, which suggests that basing eukaryotic cytotoxicity screening on haemolytic data alone is misleading [38]. If amphipathic peptide interactions with membranes, play an important role in the mechanism of action of AMPs, then membrane composition and, as a consequence, cell membrane potential probably plays a major role in contributing to differences in activities of AMPs against different bacteria and between different mammalian cell types. *E. coli* cell membranes are composed of phosphatidylethanolamine (75%), phosphatidylglycerol (20%) and cardiolipin (5%) [39,40], resulting in an overall anionic charge, with a resting membrane potential of −220 mV during early exponential phase, which increases to −140 mV in the late exponential phase [41]. In contrast, *S. aureus* membranes are composed of phosphatidylglycerol (43.1%), lysylphosphatidylglycerol (30%) and cardiolipin (22.5%) [41,42]. The electrical potential across the membrane of *S. aureus* is −130 mV at pH 7.5 [43,44], making the resting membrane potential less negative than that of *E. coli.* Likewise, there are significant differences in the membrane composition of kidney and liver cell lines. HEK293 cells have high phosphatidylcholine (~35%) and phosphatidylserine (~25%) contents and a moderate phosphatidylethanoline (~15%) content. In comparison, HepG2 cells have an outer membrane enriched in phosphatidylcholine (~30%) from which both phosphatidylethanolamine and phosphatidylserine are almost excluded [45]. As a result, both cell lines show marked differences in cell membrane potential (HepG2 cells = −18.2 mV; HEK293 cells = −40 mV) [41,42]. Having highlighted the importance of membrane composition contributing to the diversity of AMP action in different prokaryotic and eukaryotic cell membranes, it is also important to consider the other major influence, namely the role that peptide secondary structure plays in membrane–AMP interactions and therefore contributes to the biological activities of AMPs.

The amphipathicity of peptides has been widely reported as the most significant physico-chemical property of AMPs, as it enables the peptides to attack the cell membrane by interacting with the hydrophobic–hydrophilic properties of phospholipids [46]. The amphipathic nature of any peptide is determined by its secondary structure. The CD spectra for Smp24 and Smp24 S3K are very similar, suggesting a similar secondary structure between the two which, crucially, was not affected by the increase in charge on the modified peptide (+3 to +4), even though the antimicrobial activity increased. In contrast, the CD spectrum of Smp24 K7F identifies structural differences that the increased hydrophobicity introduced, leading to a peptide with reduced antimicrobial activity. It is possible that the phenylalanine residue affects the folding of Smp24 K7F into an α-helix and results in a less pronounced secondary structure. As shown in Figure 3, Smp24 K7F adopts its final helical structure at a much lower concentration of TFE than native Smp24. This may be associated with a decrease in the degree of interaction between the peptide and the hydrophobic leaflet of the membrane due to an earlier transition between the disordered and ordered states of the peptide. Furthermore, Smp24 K7F demonstrates a less defined helical structure compared with Smp24, which could be crucial for membrane insertion and antimicrobial activity.

## 4. Materials and Methods

### 4.1. Materials

Peptides were synthesised using solid-phase chemistry to 90–99% purity (David’s Biotechnologie GmbH, Regensberg, Germany). Cell lines were purchased from ATCC (Manassas, VA, USA) and routinely tested for mycoplasma. A Pierce LDH cytotoxicity assay kit was purchased from Thermo Fisher Scientific (Loughborough, UK). All other reagents, media and cell culture reagents were purchased from Sigma-Aldrich (Gillingham, UK).

### 4.2. Bacterial Strains

The bacterial isolates used in this study (*Escherichia coli* JM109, *Staphylococcus aureus* SH1000 and *Pseudomonas aeruginosa* H085180216) were obtained from the culture collection held at Sheffield Hallam University.

### 4.3. Antimicrobial Assays

All antimicrobial assays were performed by a broth microdilution method [22] in Muller Hinton (MH) broth using a CLARIOstar plate reader (BMG Labtech, Offenburg, Germany) to determine the minimum inhibitory concentration (MIC). The MIC was defined as the lowest concentration of antimicrobial peptide at which growth was inhibited when viewed as turbidity (A_600 nm_) against a negative broth control. To determine the minimum bactericidal concentration (MBC), 5 µL aliquots were taken from the wells of the MIC plate and spot-plated onto MH agar in triplicate. The plates were incubated for 18–24 h at 37 °C. The MBC was defined as the lowest concentration at which visible growth (A_600 nm_) was completely inhibited by the antimicrobial peptides [47].

### 4.4. Haemolysis Assay

Haemolytic activity was determined on sheep erythrocytes as described by Corzo, 2001 [23]. All samples were tested in triplicate and results given as HC_50_ values [24,48]. Peptides were examined in the range of 0–512 µg/mL. 10% Triton X-100 was used as a positive control and deionised water with PBS as a negative control.

### 4.5. Cytotoxicity Assays

An LDH assay (Fisher, Loughborough, UK) was used to determine the cytotoxic potential of the peptides against liver (HepG2) and kidney (HEK293) cells. Cells were grown in accordance with the manufacturer’s protocols. Cells were grown in DMEM containing 1 g/L D-glucose, 10% foetal bovine serum and 1% penicillin-streptomycin. Cells were grown to 80% confluence (37 °C, 5% CO_2_), washed twice with PBS, trypsinised, re-suspended and seeded at a density of 2.0 × 10^5^ cells/well in a 96-well plate. Cells were incubated for 24 h, and peptides or lipopeptide antibiotics (0–512 µg/mL) were added and incubated at 37 °C, 5% CO_2_ for 24 h. LDH release was determined via the manufacturer’s methods. All assays were performed in triplicate on 2 separate occasions. Triton X-100 (10%) was used as a positive control and deionised water as a negative control. Absorbance was read at 490 nm and 680 nm on a plate reader (CLARIOstar). LD_50_ values were calculated by plotting a dose–response curve and determining the dosage that would cause 50% cell death.

### 4.6. Therapeutic Indices

The therapeutic indices of Smp24 and the modified peptides were calculated by dividing the LD_50_ and HC_50_ values by the MIC for each peptide against *E. coli* [25].

### 4.7. Circular Dichroism (CD) Spectra

CD spectra were recorded on a JASCO J-810 spectropolarimeter at 22 °C using a 0.1 cm path length cell over a spectral range of 190–250 nm. Peptides were dissolved in water to a 100 μM concentration. Aqueous 2,2,2-trifluoroethanol (TFE) was titrated to obtain a final concentration of 60% TFE. Samples were tested in triplicate.

### 4.8. Modelling Studies

#### 4.8.1. Helical Wheel Projections

Helical wheel projections were produced using the Pepwheel software provided by the European Molecular Biology Organization (https://emboss.bioinformatics.nl/cgi-bin/emboss/pepwheel, accessed on 16 June 2022).

#### 4.8.2. Ab-Initio Modelling

Protein secondary structure prediction was carried out using either Jpred3 [49] or Quark (https://zhanglab.ccmb.med.umich.edu/QUARK/, accessed on 16 June 2022) online servers.

#### 4.8.3. 3D Modelling

3D modelling was achieved using PyMOL 2.3 (https://pymol.org/2/, accessed on 16 June 2022).

#### 4.8.4. Hydrophobic Moment

The Hydrophobic Moment Vector Calculator (3D-HM) [28], was used to characterize surface polarity (https://www.ibg.kit.edu/HM/?page=index, accessed on 16 June 2022).

## 5. Conclusions

In conclusion, this study has demonstrated that in an amphipathic AMP, the balance between hydrophobic domains and charged domains is crucial, and biological activity can be easily reduced if this balance shifted too much in either direction. The biological activity of Smp24 may be enhanced or reduced by performing single amino acid substitutions. Introducing a single positive charge near the N-terminus (Smp24 S3K) has provided a modified peptide with a markedly improved TI against both liver and kidney cell lines. The D23F modification produced a significantly improved TI over the parent peptide in liver cells but had a decreased TI against erythrocytes, which highlights the importance of a comprehensive cytotoxicity screen when investigating the properties of AMPs in order to evaluate a TI.

## Figures and Tables

**Figure 1 ijms-23-07979-f001:**
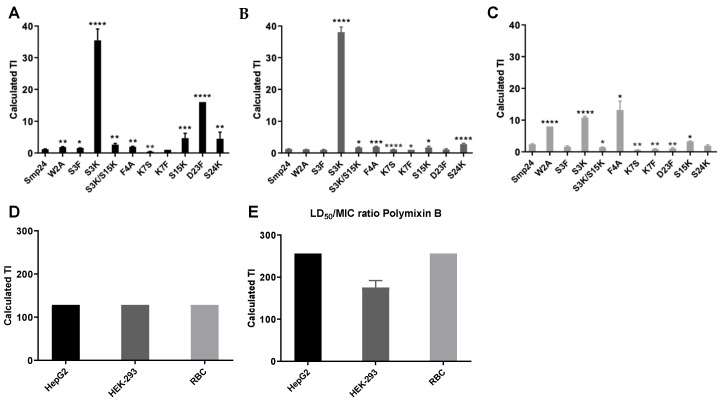
Therapeutic indices of Smp24 and derived modifications. TI calculations of peptides against HepG2 (**A**), HEK293 (**B**), sheep erythrocytes (**C**), daptomycin against all cell lines (**D**) and polymyxin B against all cell lines (**E**). TIs of AMPs and PMB were calculated against *E. coli*. As daptomycin is only effective against Gram-positive isolates, its TI was calculated against S. aureus. Statistical significance of the difference between Smp24 and its derivatives was determined by performing a *T*-test on TI data and denoted by * equal to *p*-value < 0.05, ** *p*-value < 0.01, *** *p*-value < 0.001 and **** *p*-value < 0.0001.

**Figure 2 ijms-23-07979-f002:**
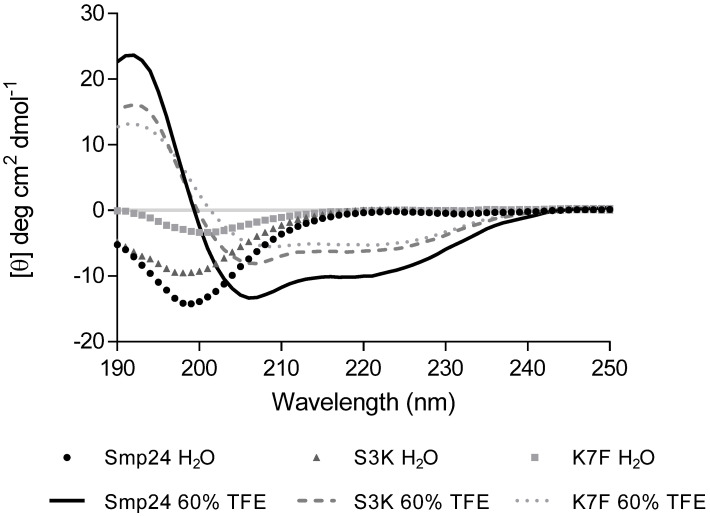
Solvent-dependent effects of Smp24, Smp24 S3K and Smp24 K7F on peptide secondary structure determined by CD.

**Figure 3 ijms-23-07979-f003:**
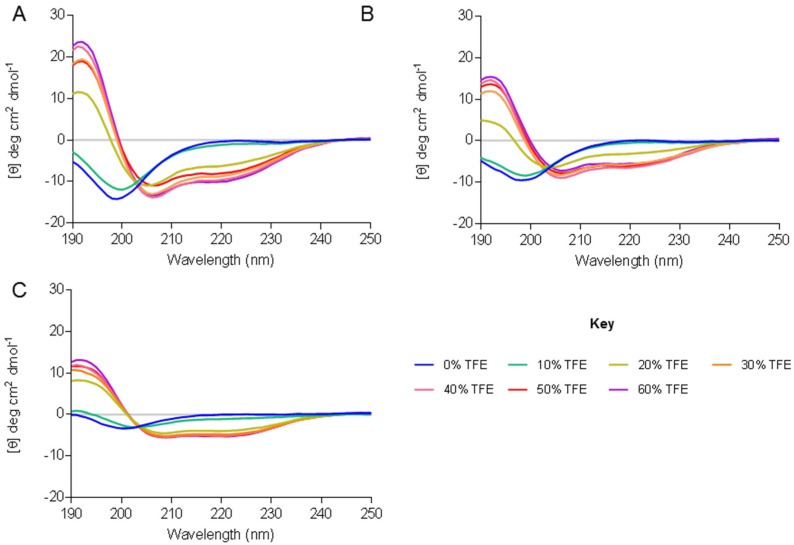
Titration of TFE and its effect on secondary structure. TFE was titrated from 0% to achieve a final concentration of 60% TFE. CD spectral traces for Smp24 (**A**), Smp24 S3K (**B**) and Smp24 K7F (**C**).

**Figure 4 ijms-23-07979-f004:**
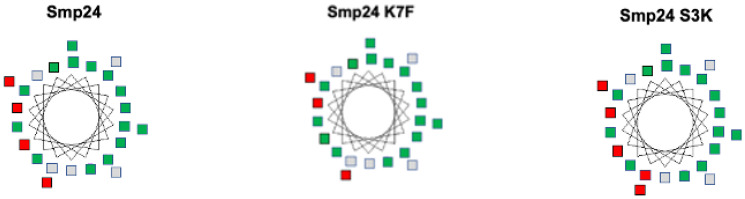
Helical wheel analysis of Smp24, Smp24 K7F and Smp24 S3K. Charged residues are red and hydrophobic non-polar residues are green.

**Figure 5 ijms-23-07979-f005:**
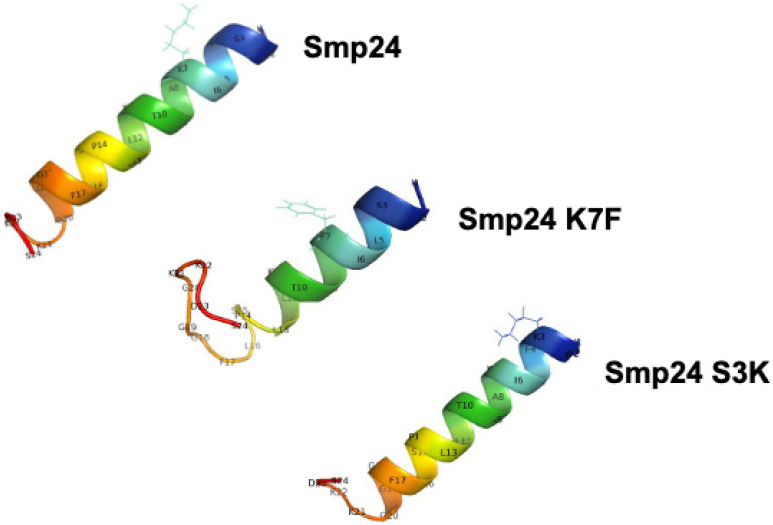
Secondary structures of Smp24 peptides visualized by PyMOL.

**Figure 6 ijms-23-07979-f006:**
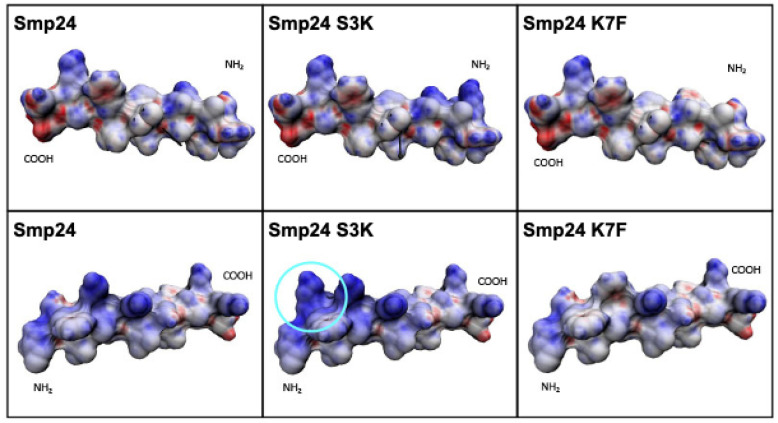
Space filling models of Smp24 and Smp24 derived peptides. Polar facets are highlighted in blue and non-polar facets are highlighted in red, using the HM calculator [28].

**Table 1 ijms-23-07979-t001:** Sequences of peptides used in this study. Amino acids that differ from the Smp24 peptide are in bold.

Peptide	Sequence
Smp24	IWSFLIKAATKLLPSLFGGGKKDS
Smp24 W2A	I**A**SFLIKAATKLLPSLFGGGKKDS
Smp24 S3F	IW**F**FLIKAATKLLPSLFGGGKKDS
Smp24 S3K	IW**K**FLIKAATKLLPSLFGGGKKDS
Smp24 S3K/S15K	IW**K**FLIKAATKLLP**K**LFGGGKKDS
Smp24 F4A	IWS**A**LIKAATKLLPSLFGGGKKDS
Smp24 K7S	IWSFLI**S**AATKLLPSLFGGGKKDS
Smp24 K7F	IWSFLI**F**AATKLLPSLFGGGKKDS
Smp24 S15K	IWSFLIKAATKLLP**K**LFGGGKKDS
Smp24 D23F	IWSFLIKAATKLLPSLFGGGKK**F**S
Smp24 S24K	IWSFLIKAATKLLPSLFGGGKKD**K**
Daptomycin	WDNDTGODDADGD
Polymyxin B	DabT(Dab)_3_FL(Dab)_2_T

**Table 2 ijms-23-07979-t002:** MIC and MBC of Smp24 and modified peptides against a range of bacterial strains. Overall charge and hydrophobicity (grand average of hydropathicity index (GRAVY)) [21,16] are also shown. nc denotes not calculable due to the mixed population of the two separate polymyxins.

Peptide	Charge	GRAVY	MIC (mg/L)			MBC (mg/L)		
			*E. coli*	*P. aeruginosa*	*S. aureus*	*E. coli*	*P. aeruginosa*	*S. aureus*
Smp24	+3	0.31	32	64	8	32	64	8
Smp24 W2A	+3	0.43	64	128	32	64	256	64
Smp24 S3F	+3	0.46	32	64	16	64	128	64
Smp24 S3K	+4	0.18	1	32	0.5	8	128	1
Smp24 S3K/S15K	+5	0.05	16	16	16	16	32	16
Smp24 F4A	+3	0.27	32	128	16	64	256	16
Smp24 K7S	+2	0.44	64	128	16	128	256	32
Smp24 K7F	+2	0.59	>512	>512	>512	>512	>512	>512
Smp24 S15K	+4	0.18	8	32	4	8	32	8
Smp24 D23F	+4	0.58	32	64	32	32	64	32
Smp24 S24K	+4	0.18	8	32	4	8	128	8
Daptomycin	−6	−1.93	>512	>512	4	>512	>512	8
Polymyxin B	+5	nc	2	2	32	4	8	64

**Table 3 ijms-23-07979-t003:** Haemolytic activity of Smp24 and its derivatives. Statistical significance of the difference between Smp24 and its derivatives was determined by performing a Student’s *T*-test on the data. * denotes *p*-value < 0.05. nc denotes not calculable due to the mixed population of the two separate polymyxins.

Peptide	Charge	GRAVY	HC_50_ (μg/mL)
Smp24	+3	0.31	76
Smp24 W2A	+3	0.43	>512 *
Smp24 S3F	+3	0.46	52
Smp24 S3K	+4	0.18	11 *
Smp24 S3K/S15K	+5	0.05	23 *
Smp24 F4A	+3	0.27	423 *
Smp24 K7S	+2	0.44	36
Smp24 K7F	+2	0.59	442 *
Smp24 S15K	+4	0.18	26 *
Smp24 D23F	+4	0.58	33 *
Smp24 S24K	+4	0.18	15 *
Daptomycin	−6	−1.93	>512
Polymyxin B	+5	nc	>512

**Table 4 ijms-23-07979-t004:** Cytotoxicity profiles of Smp24 and its derivatives against human kidney (HEK293) and liver (HepG2) cells. Statistically significant increases or decreases in cytotoxic profile between Smp24 and its derivatives are labelled * denotes *p*-value < 0.05. nc denotes not calculable due to the mixed population of the two separate polymyxins.

Peptide	Charge	GRAVY	LD_50_ (μg/mL)	
			HepG2	HEK293
Smp24	+3	0.31	37	39
Smp24 W2A	+3	0.43	121 *	72 *
Smp24 S3F	+3	0.46	51 *	32
Smp24 S3K	+4	0.18	34	38
Smp24 S3K/S15K	+5	0.05	42	29 *
Smp24 F4A	+3	0.27	63 *	56 *
Smp24 K7S	+2	0.44	30	15 *
Smp24 K7F	+2	0.59	>512 *	>512 *
Smp24 S15K	+4	0.18	38	13 *
Smp24 D23F	+4	0.58	>512 *	38
Smp24 S24K	+4	0.18	36	23 *
Daptomycin	−6	−1.93	>512	>512
Polymyxin B	+5	nc	>512	351

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
