# Peer review of "Improving the Therapeutic Index of Smp24, a Venom-Derived Antimicrobial Peptide: Increased Activity against Gram-Negative Bacteria"

_ijms, 2022, doi:10.3390/ijms23147979_

Round 1
Reviewer 1 Report
In the paper entitled "Improving the therapeutic index of Smp24, a venom derived antimicrobial peptide: increased activity against Gram-negative bacteria", the authors present the structural modulation (aminoacid sequence) of AMP Smp24 in order to improve its therapeutic index.
Considering the rising interest for AMP and for peptides in general, I find the idea of this research a very interesting one. Also, the research is very complex, the paper is well organised and written and suitable for publication in IJMS, after a minor revision:
- all abbreviations should be defined when first used within the text;
- there is no methodology regarding the peptide synthesis (in solution, SPPS, characterization); this information must be provided.
Author Response
We thank the referee for their comments.
- all abbreviations should be defined when first used within the text;
We apologise for this oversight, we have now checked the entirety of the document and all abbreviations are now defined when first used and the abbreviated form is now used from that point forward.
- there is no methodology regarding the peptide synthesis (in solution, SPPS, characterization); this information must be provided.
We apologise, we have included the information that the peptides were synthesised using solid phase chemistry in the materials section (line 398).
Reviewer 2 Report
I have carefully read the manuscript by Rawson et al, entitled “Improving the therapeutic index of Smp24, a venom derived antimicrobial peptide: increased activity against Gram-negative bacteria”. In my opinion, the manuscript falls in the IJMS journal’s scope, the introduction is well written and the experiments are well conducted. However, I suggest some modifications before possible publication in IJMS journal, as reported below:
-Page 1, lines 40-42: this part is poor and must be revised. Indeed, the authors can include the principal organisms which are a source of antimicrobial peptides. For example, AMPs are found several organisms like fungi/mushrooms. Recently, an updated review of bioactive peptides from mushrooms, describe also the presence of AMPs in these organisms. The same also for plant AMPs and so on. On this regard, I suggest to revise the bibliography accordingly, including information supported by appropriate references;
-Table 1: The bold is ok, but I suggest also to underline the amino acid substitutions, to find them more quickly;
-Page 3, line 88: use the full name (scientific name in Italics) for the bacterial species the first time it appears, then use the abbreviation;
-Page 3, line 96: change ‘μg/ml’ by ‘μg/mL’. Check this in overall manuscript and Tables;
-Page 5, lines 149-150 (Figure legend): use the italics for the scientific names. Check this in overall manuscript and Figure legends.
Author Response
We thank the reviewer for their comments and I will address each point raised below:
-Page 1, lines 40-42: this part is poor and must be revised. Indeed, the authors can include the principal organisms which are a source of antimicrobial peptides. For example, AMPs are found several organisms like fungi/mushrooms. Recently, an updated review of bioactive peptides from mushrooms, describe also the presence of AMPs in these organisms. The same also for plant AMPs and so on. On this regard, I suggest to revise the bibliography accordingly, including information supported by appropriate references;
We agree that we could have been more comprehensive in our description of the breadth of organisms that produce AMPs. We have revised the paragraph and have included an additional 8 citations covering AMPs from bacterial, archaebacterial, protozoan, fungal, plant and animal sources including specific examples (lines 42-49).
-Table 1: The bold is ok, but I suggest also to underline the amino acid substitutions, to find them more quickly;
We agree with the reviewer that underlining the substituted residue in addition to bolding it makes it much clearer for the reader. We have changed this throughout Table 1 (lines 101-102).
-Page 3, line 88: use the full name (scientific name in Italics) for the bacterial species the first time it appears, then use the abbreviation;
We apologise for this oversight and have corrected the error and checked the rest of the document.
-Page 3, line 96: change ‘μg/ml’ by ‘μg/mL’. Check this in overall manuscript and Tables;
We have changed ‘μg/ml’ to ‘μg/mL’ throughout the document.
-Page 5, lines 149-150 (Figure legend): use the italics for the scientific names. Check this in overall manuscript and Figure legends.
Again, we apologise for the oversight and have resolved this throughout the document.
Reviewer 3 Report
In their manuscript, Rawson et al. describe the modification of an anti-microbial peptide from the scorpion Scorpio maurus palmatus by exchanging certain amino acids and testing the anti-bacterial activity against E. coli, P. aeruginosa and S. aureus as well as the hemolytic and cytotoxic activity against mammalian cells. Two of such variants they further characterize structurally by CD spectroscopy as well as in silico analyses. They discover a variant that exhibits improved properties. The manuscript is well written. The work is well carried out and derives interesting results. Anti-microbial peptides represent a competitive subject of investigations since they offer the possibility to be applied in case of infections instead of antibiotics. Development of bacterial resistance against such compounds is much less likely compared to antibiotics. Therefore, the manuscript should be published. Two very minor point need to be addressed:
1. Introduction line 42: here it is claimed that: AMPs protect the animal from exchange of body fluids when stinging prey. This sounds odd and needs some short explanation if correct.
2. Figure 1: B is missing from panel B.
Author Response
We thank the referee for their comments.
Introduction line 42: here it is claimed that: AMPs protect the animal from exchange of body fluids when stinging prey. This sounds odd and needs some short explanation if correct.
We apologise for our clumsy turn of phrase in this sentence, we have revised the sentence to make the interaction, and resultant infection risk, between the scorpion and their prey more clear and have included a supporting citation (lines 47-49).
Figure 1: B is missing from panel B.
We apologise, this must have happened during the pdf conversion and I must have missed it when I reviewed the proof. On our Word document the B is visible in the panel. I will make sure that when the updated version of the manuscript is submitted and converted that I will look more closely at the proof.